# The Patterns and Impact of Off-Working Hours, Weekends and Seasonal Admissions of Patients with Major Trauma in a Level 1 Trauma Center

**DOI:** 10.3390/ijerph18168542

**Published:** 2021-08-12

**Authors:** Husham Abdelrahman, Hassan Al-Thani, Naushad Ahmad Khan, Monira Mollazehi, Mohammad Asim, Ayman El-Menyar

**Affiliations:** 1Department of Surgery, Trauma Surgery, Hamad General Hospital, Doha P.O. Box 3050, Qatar; hushamco@hotmail.com (H.A.); althanih@hotmail.com (H.A.-T.); 2Clinical Research, Trauma and Vascular Surgery, Hamad General Hospital, Doha P.O. Box 3050, Qatar; naushadkhan82@gmail.com (N.A.K.); masim1@hamad.qa (M.A.); 3Qatar National Trauma Registry, Trauma Surgery, Hamad General Hospital, Doha P.O. Box 3050, Qatar; mollazehi@hamad.qa; 4Department of Clinical Medicine, Weill Cornell Medical College, Doha P.O. Box 24144, Qatar

**Keywords:** trauma, weekend effect, off-working hours, trauma systems, injury mechanism, injury severity, Qatar

## Abstract

Background: The trauma incidence follows specific patterns in different societies and is expected to increase over the weekend and nighttime. We aimed to explore and analyze the incidence, pattern, and severity of trauma at different times (working hours vs. out off-working hours, weekdays vs. weekends and season). Methods: A retrospective analysis was conducted at a level 1 trauma facility in Qatar. All injured patients admitted between June 2017 and May 2018 were included. The data were analyzed to determine whether outcomes and care parameters of these patients differed between regular working hours and off-working hours, weekdays vs. weekends, and between season intervals. Results: During the study period, 2477 patients were admitted. A total of 816 (32.9%) patients presented during working hours and 1500 (60.6%) during off-working hours. Off-working hours presentations differed significantly with the injury severity score (ISS) (*p*  <  0.001), ICU length of stay (*p*  =  0.001), blood transfusions (*p* = 0.001), intubations (*p* = 0.001), mortality rate (9.7% vs. 0.7%; *p* < 0.001), and disposition to rehabilitation centers. Weekend presentations were significantly associated with a higher ISS (*p* = 0.01), Priority 1 trauma activation (19.1% vs. 14.7%; *p* = 0005), and need for intubation (21% vs. 16%; *p* = 0.002). The length of stay (ICU and hospital), mortality, and disposition to rehabilitation centers and other clinical parameters did not show any significant differences. No significant seasonal variation was observed in terms of admissions at the trauma center. Conclusions: The off-working hours admission showed an apparent demographic effect in involved mechanisms, injury severity, and trauma activations, while outcomes, especially the mortality rate, were significantly different during nights but not during the weekends. The only observed seasonal effect was a decrease in the number of admissions during the summer break.

## 1. Introduction

Trauma is a leading cause of death, morbidity, and permanent disability worldwide [1]. The incidence of trauma is rising [1]. The trauma incidence follows specific patterns in different societies. The severity of injuries and outcomes may be affected by the time of day, the days of the week, and the seasons [2,3]. It is plausible to assume that disparities in a population’s social and economic activities and their general lifestyle have a differential impact on the occurrence and pattern of injuries, primarily on hospital resource utilization [4]. These factors, in turn, may raise the likelihood of injury and a poor outcome. Most of the research on the relationship between time and injury focuses on the impact of time of admission on injury’s clinical outcomes, notably mortality and hospital length of stay.

During off-hours, most medical institutions drastically reduce employee levels [5]. Additionally, less hospital staffing, particularly the seniors and certain investigational services, may not be immediately available during off-hours and weekends. Off-working hours presentation are thus known to be a risk factor for patients presenting with unexpected urgent conditions requiring immediate diagnosis and prompt action [6]. On the other hand, limited research indicates that the mortality of the injured patients treated at a level 1 trauma center is the same whether admitted during off-working hours or regular working hours [2,7,8,9,10]. Level 1 trauma centers have the highest concentration of medical resources and personnel availability across the week [11,12].

Furthermore, few available studies, mostly descriptive, have shown inconsistencies with a wide range of results during weekends, weekdays, and nights, defining these “off hours” timeframes in the literature problematic. This likelihood for the increased trauma admissions “off-working hours” cross over to a never-ending debate and ongoing uncertainty, and a compelling, often unanswered question: “is the degree of care provided to a patient hospitalized after working hours and on weekends the same as that provided to a patient admitted during the week?” [2,3,12,13].

For example, specific system-based weekend admission observations were described by Bell et al.; admissions during weekends are restricted overall (i.e., fewer patients) and for the sicker patients [12]. This question about the level of care is not new and has been reported by different specialities in different patient populations [2]. For example, historical differences in the mortality of babies born over the weekend exist even in developed countries such as the USA, UK, and Australia [12,13].

A separate though related question also arises: “do specific weather conditions or seasons impact trauma admissions and outcomes?” These details potentially influence care, resource availability, optimization, and trauma patients’ outcomes and should be investigated to guide and tailor approaches to care during vulnerable times [6,13,14,15,16]. Furthermore, differences in outcomes, the length of hospital stay, and death rates exist among hospitals and emergency admissions during different weekdays [17,18]. Higher mortality on weekends and nights than weekdays is well documented [19,20].

There has been no conclusive response to whether the “off-working hours impact” is a truth or a myth among healthcare practitioners. The problem is that individual studies show a relative increase in mortality rates rather than an absolute increase in fatalities or that some earlier analyses misread databases rather than proving “bad” medical treatment. Bell and Redelmeier compared hospital mortality between patients admitted on the weekend vs. weekday for three acute medical conditions and concluded that weekend admissions were associated with significantly higher mortality than weekday admissions, even after adjusting for age, sex, and coexisting conditions [12]. An earlier study showed that once the trauma admission and demographic patterns have been identified, the utilization, modification, and design of the trauma system would be appropriately planned and implemented [17]. Moreover, another report concluded that measuring the ‘off-hours’ or ‘weekend’ effect on trauma mortality could be considered as a quality indicator that reflects the outcome as well as the process indicator [21].

The Reviewing Equitable Access to Health Care outcomes out of Hours and at the weekends (REACH) project is a milestone analysis. The report found mixed evidence of inequitable outcomes for patients admitted during off-working hours, based predominantly on the retrospective analysis of large administrative datasets. On the other hand, severity-adjusted clinical dataset analyses are less supportive that differential outcomes do exist [22].

Thus, there are several gaps in the literature which include a lack of answers to the questions stated above and the fact that the answers may be system related. Carrying such an analysis in each country and maybe in each specialty is vital. Considering this, we sought to investigate and analyze the impact of time of injury to gain insight into the effect of hours of the day and days of the week or seasonal patterns on injury characteristics, incidence, outcomes, and the potential level of care provided to trauma patients. This could have an implication on the injury prevention settings and preparedness. To the best of our knowledge, data in this regard are lacking in our region.

## 2. Materials and Methods

### 2.1. Study Design and Population and Setting

This study is a retrospective analysis of trauma registry data. The period of the study was one year, starting from June 2017 to the end of May 2018. The database included all admitted trauma patients to Hamad Trauma Centre (HTC). HTC is a level 1 accredited trauma center that is a consultant lead service; a multidisciplinary team that provides care 24 h seven days a week with an advanced American college of surgeon equivalent level of care (trauma field triage for referral to the center, inter-facility transfer protocols shared by pre-hospital providers (ambulance service), immediate trauma surgeon availability, high standard for care (resuscitation, massive transfusion protocol (MTP), diagnostic access (point of care (arterial blood gas (ABG), rapid international normalizing ration (INR), thromboelastogram (TEG)/(ROTEM)), emergency laboratory, imaging focused assessment sonogram of trauma (FAST)/extended FAST/pan-computed tomography (P-CT), and other selective imaging and MRI), access to a trauma intensive care unit (TICU), and immediate access to the operating theatre (OT). The trauma system is a Canada-accredited system defined as an organized, coordinated effort in a specific geographic area that provides a full range of care to all injured patients while also being integrated into the local public health system [7]. All admitted patients who died at ED were excluded from the study. Ethical approval for the study was granted from the medical research center and institutional review board of Hamad Medical Corporation, Doha, Qatar (MRC-01-18-432).

### 2.2. Exposures, Variables, and Outcome Measures

Demographic information, time of hospital admission (hour and day of the week), injury features, and mode of arrival to ED, hospital resource use, and discharge disposition were included in the data. Weekends (weekend-days and weekend-nights) and weekdays were the two exposure variables. Weekdays were defined as the hours (h) between 06:01 and 18:59 h Sunday through Thursday, and weeknights as the hours between 15:00 and 06:00 h. It is vital to note that the weekday in the state of Qatar begins on Sunday and finishes on Thursday and the weekends are Fridays and Saturdays (starting from 6:00 h on Friday and lasting till 5:59 h on Sunday).

We arbitrarily chose 6:01 h and 15:00 h as the cut-off timings as these are the regular working hours for staff in our hospital. Working hours admissions are defined as any admission between 06:01 and 15:00 h, while off-working hours admissions include any admission after that until the following morning (15:01–06:00 h)—the rationale behind choosing these times was to incorporate all night admissions throughout the year with the off hours to compensate for a 2 h difference during the year. The breakdown of admission was carried out according to seasonal periods: June–August, September–November, December–February, March–May.

We examined the clinical features, trauma care, and outcomes of injured patients requiring specialized treatment presented outside normal business hours to those presented within regular business hours. The primary exposures were ED admissions time, day, and seasonal presentations. Management indicators and clinical outcomes were analyzed in relation to time (working hours vs. off-working hours, weekdays vs. weekends, and seasons). The trauma team includes the following: consultant on call for T1 activation which is available within 15 min of trauma activation regardless of the timing of the day or weekday. Additionally, 2 specialists covering 4 shifts (in-house) over 24 h across the 7 days of the week. The residents are rotating to accompany the specialist across all the shifts.

Data variables included age, sex, and injury severity as represented by the injury severity score (ISS), vital signs measured immediately after admission to the emergency room (ER), the injury type, and mechanism of injury. Trauma care parameters were pre-hospital time (time from emergency call to ER arrival), ER stay time (time from ER arrival to OR), and total time to OR (time from emergency call to OR), scene time, and hospital management. Admission time was defined as the time the patient arrived in the trauma resuscitation unit (TRU). The process measure included: standard trauma team activation, advanced trauma life support (ATLS) guidelines, intubation, ventilation, FAST, computed tomography (CT) scans, images, operation, TICU. The standard trauma team activation is based on the EMS dispatch system, which is on call for service code into the patient’s priority or P1 (emergency), P2 (urgent), and others (routine). The response time (RT) was estimated as the duration between the time a call was made for the emergency and the arrival of the ambulance at the emergency scene.

### 2.3. Statistical Analysis

When applicable, data were presented as a proportion, mean (standard deviation), median, and range or interquartile range. The chi-squared test was performed to compare proportions between categorical groupings for each subgroup of patients. The Kolmogorov–Smirnov test checked the normality of continuous variables. For parametric data, continuous variables were compared using the Student’s *t*-test for two groups or the one-way ANOVA test for more than two groups. Yates’ adjusted chi-square was employed for categorical variables if the anticipated cell frequencies were less than five. Data were compared in different ways (admissions on weekdays vs. weekend, admissions on off-working h vs. working h, December–February vs. March–May, June–August, and September–November, survivors vs. non-survivors). Multivariate logistic regression analysis was performed for the predictors of mortality after adjusting for the most relevant covariates. Data were expressed as the odds ratio and 95% confidence interval. Statistical significance was defined as a two-tailed *p*-value of less than 0.05. SPSS version 21 was used to analyze the data (SPSS Inc., Chicago, IL, USA). IBM SPSS Statistics for Windows, Version 21.0, was used for all statistical analyses (IBM Corp., Armonk, NY, USA).

## 3. Results

During the study period, 2477 patients were admitted to the HTC database. The patients’ mean age was 30.9 ± 15.8 years. The majority (84%) were men, the majority (79%) were expats from 33 different countries, and only 21% were Qataris. The most common mechanism of trauma was a road traffic accident (49%) followed by a fall from height (28%) and the fall of heavy objects (5%). Sixteen percent of patients sustained severe trauma that necessitated level 1 activation (P1), which required more hospital resources and poor prognosis. Overall, 169 (6.8%) patients died in our study. Regarding admissions, 816 (32.9%) patients presented during working hours while 1500 (60.6%) presented during off-working hours. A total of 1749 (1749) patients presented during weekdays, while the rest (728 patients) were admitted during weekends (Figure 1). The majority (almost 58%) of the patients were presented to the ED during 03:01–6:00 h (32.3%) and 00:01–3:00 h (26.3%). A significant percentage of patients present in the early morning hours (dawn), which is the time workers commute to their workplace, and during the early afternoon, which corresponds to the return from the working shift (Figure 2). The weekdays vs. weekends and seasonal breakdown are shown in the Figure 3a,b. Proportion-wise, most of the patients were admitted to ED during the weekend (Fridays and Saturdays). Additionally, the majority (almost 58%) of the patients were presented to the ED during 03:01–6:00 and 00:01–3:00 h during weekends and on weekdays (Figure 3a).

The seasonal breakdown of ED admissions is shown in Figure 3b: June–August, 547 patients; September–November, 639 patients; December–February, 652 patients; and March–May, 639 patients. The number of admissions dropped significantly during the summer season, which is the school break time in the state of Qatar.

### 3.1. Comparison of Patients Admitted during Working Hours and Off-Working Hours (Off-Hours Effect)

Table 1 presents the comparison of demographics, clinical characteristics care, and outcomes between working hours and off-working hours in a day. The off-working hours’ patients were younger than patients presented during regular working hours (mean 28.8 ± 15.5 vs. 33.5 ± 15.3; *p* < 0.001); 79.8% were 16–64 y, 17.9% were ≤15 y, and only 2.2% were ≥65 y. The proportion of off-working-hours females presented during off-working hours was higher compared to those presented during regular working hours (17% vs. 13%, respectively). Regarding the mechanism of injury, road traffic-related injuries (RTI) were the most common (53%), followed by falls (29.5%) during working hours. The same pattern was observed for off-working hours presentations (47.5% RTI and 27.3% falls). More males (87%) presented during working hours than during the off-working hours. Pre-hospital time was significantly shorter during off-working hours than during regular working hours presentations (*p* = 0.03).

Patients admitted during off-working hours were significantly younger and had a significantly higher ISS (14.8 ± 11.6 vs. 11.2 ± 7.8; *p* < 0.001) and longer ICU length of stay (LOS) (*p*  =  0.001) than those admitted during regular working hours. No differences were observed with respect to hospital LOS between the off-working hours and working hours presentations (*p* = 0.70). A significant difference was observed for blood transfusion (17.1% vs. 8.7%; *p* = 0.001) and intubation (24.4% vs. 5.0%; *p* = 0.001) between off-working hours and working hours presentations. The mortality rate (9.7% vs. 0.7%; *p* < 0.001) and disposition to rehabilitation centers (6.2% vs. 3.8%; *p* = 0.01) of off-working hours presentation was significantly higher than during regular working hours (Table 1). Overall, the patients with off-working hours presentations had a significantly shorter pre-hospital time, higher use of blood transfusion, massive transfusion protocol activation, more intubations, early surgeries, and exploratory laparotomy, suggestive of bleeding-related indications for surgery.

### 3.2. Comparison of Patients Admitted on Weekdays and Weekends (Weekend’s Effect)

Table 2 shows the comparisons of demographics, admissions, injury severities, the median length of stay, mortality, and other care-related variables for patients admitted on weekends vs. weekdays. Higher proportion of the patients were admitted on the weekend compared to a regular working day (if we divide the number of patients by the number of days (1749 patients and 5 working days), then there are around 350 patients/day, while weekend daily admissions (728 patients and 2 days) have around 364/day; without this consideration, there would not be a weekend number effect.)

There was no statistical difference concerning the age or mechanism of injury. The proportion of P1 activation was higher in patients presented on weekends than on weekdays (19.1% vs. 14.7%; *p* = 0005). Additionally, weekends showed a higher admission rate for females (18% vs. 14.6%) and a lower admission rate for males (82% vs. 85%) in comparison to the weekdays (*p* = 0.02). Furthermore, shorter pre-hospital time less than or equal to 60 min during the weekends (37%) was observed compared to the weekdays (31%) (*p* = 0.01).

More patients needed intubation during the weekends (21% vs. 16%; *p* = 0.002). The ISS was significantly higher during weekends than during weekdays (*p* = 0.01), but other clinical characteristics and care and outcome parameters did not differ between the two groups. No significant differences were noticed in blood transfusion, MTP activation, early surgery, or exploratory laparotomy. The length of stay (ICU and hospital), mortality, and disposition to rehabilitation centers did not show any significant differences.

### 3.3. Comparison of Patients Admitted to Hospital Stratified by Seasons (Season’s Effect)

Table 3 depicts the comparisons of demographics, admissions, injury severities, the median length of stay, mortality, and other care-related variables for patients admitted on stratified by seasons. No significant variation in the number of trauma admissions across the four seasons was observed (*p* = 0.29). Most admissions occurred from December to February (*n* = 652), and the fewest occurred in the summer (June to August; *n* = 547). It should be noted that the summer season in Qatar involves school breaks, which could be the likely reason for the relatively lesser number of patients compared to the rest of the seasons.

No significant seasonal variation was observed in terms of admissions at the trauma center. The number of patients, mean age, breakdown of age ranges, gender, and mechanism of the injury did not reflect any statistical difference for the different season’s intervals year. The type of trauma activations (P1, P2, others) and the pre-hospital times did not show any difference among the four seasons of the year. Blood transfusion, MTP activation, intubation, early operations, and the requirement for exploratory laparotomies all had no statistically significant differences. There was no significant difference in the mean ISS, median ICU LOS, hospital LOS, and disposition to rehabilitation centers by season (*p* = 0.59, 0.19, 0.70, and 0.30, respectively). Similarly, there was no significant difference in the number or proportion of mortality stratified by seasons in our study (*p* = 0.34).

### 3.4. Survivors versus Non-Survivors

Table 4 shows the comparisons of different variables among survivors and non-survivors. The survivors were significantly younger (30.8 ± 15.8 vs. 34.4 ± 16.0; *p* = 0.03) and were predominantly males (*p* = 0.001). There was no statistical difference concerning the prehospital time, OR within 24 h, patients presented at ED on weekdays and season between the two groups. In comparison to survivors, those who died were more likely to be injured by RTI (OR 2.76 (1.73–4.39, *p* = 0.001), had frequent head injuries (5.72 (4.13–7.92), *p* = 0.001), required P1 activation (OR 2.06 (1.45–2.92), *p* = 0.001), MTP activation (OR 13.10 (8.29–20.69), *p* = 0.001), and had significantly higher ISS (22.9 ± 15.7 vs. 12.3 ± 9.1, *p* = 0.001). Non-survivors were more likely to be admitted during off-working hours than survivors (14.46 (6.36–32.88), *p* = 0.001).

### 3.5. Predictors of Mortality in Trauma Patients

Table 5 shows the findings of the multivariate regression analysis to look for the independent predictors of mortality. After adjusting for the potential confounders, age (aOR 1.020 (95% CI 1.004–1.035), *p* = 0.01), injury severity score (aOR 1.042 (95% CI 1.018–1.067), *p* = 0.001), head injury (aOR 4.152 (95% CI 2.310–7.464), *p* = 0.001), MTP activation (aOR 4.192 (95% CI 2.127–8.262), *p* = 0.001), and ED presentation during off-working hours (aOR 6.329 (95% CI 2.673–14.985), *p* = 0.001) were found to be the significant predictors of mortality.

## 4. Discussion

This is a unique study in the Arab Middle Eastern region that describes the patterns and impact of different timings of admissions of trauma patients in a level 1 trauma center. Efforts are being undertaken to improve healthcare delivery to reduce the gaps in healthcare outcomes, regardless of whether patients are admitted during the week, at night, or on weekends. In the current study, the research objective was to uncover whether there are unequal health outcomes for trauma patients admitted outside of working hours vs. regular working hours, weekdays versus weekends, and in varying seasons of the year in Qatar. We tested the hypothesis of whether the care and outcomes of injured patients requiring specialized intervention at a tertiary care level 1 trauma center in Qatar differed significantly during off-working hours versus regular working hours. In our study, patients admitted during off-working hours were younger and had a significantly higher ISS, longer ICU LOS, and higher mortality rate than those presented during regular working hours. Our results corroborated with the observation of Mitra et al. [23] (Australia, 2014). A retrospective clinical investigation found that off-hours care was related to poor outcomes in critically ill trauma patients [23]. They observed that patients who presented after hours had a much greater mortality rate than those presented during regular hours (43.1% vs. 33.1%). Another researcher reported a similar observation: Di Bartolomeo et al. [21] (2014, Italy) observed that night-time admission had a substantial negative influence on mortality (OR = 1.49; 95% CI 1.05 to 2.11). Overall, the patients with off-working hours presentations required frequent intubations, suggestive of severe trauma. This finding might have some plausible explanation. As critically injured patients require the assistance of multiple appropriately qualified caregivers simultaneously, the lower staffing level during off-working hours may have negatively affected patient management performance. Our findings underscore the need for emergency care professionals to be aware of this disadvantage and, as a result, to be more alert about the risk of intubation in ED patients treated during off-working hours and weekends.

However, in contrast to our and the abovementioned findings, several previous studies have reported that off-working hours presentation did not adversely affect survival and unexpected outcomes in trauma patients [7,8,9,24]. In a retrospective cohort study of clinical data at a level 1 trauma center (2005, USA), Arbabi et al. [7] discovered that no difference in mortality or LOS was detected in conjunction with weekend or LOS after correcting for confounding variables of night admission. Another retrospective study by Busse et al. (2004, USA) reported that the time of presentation (nighttime vs. daytime weekend vs. weekday, the month of the year, and year) was not associated with in-hospital mortality after adjustment of illness and other confounding variables [9]. Metcalfe et al. (UK) [3] and others [12,25,26] did not find the effects of the time of day or day of the week on injured patients. Ono et al. (2015, Japan) reported that off-working hours presentation was associated with longer ER stays for patients ISS >15. Off-working hours presentation was also associated with an increased risk of adverse events in the ED. After adjustment for confounders, no differences were detected in mortality and unexpected death between off-working hours and regular working hours [27].

Interestingly, a more recent national-level Japanese study by Hirose et al. (2020, Japan) [28] evaluated the outcomes of trauma patients according to the time of day or day of the week of emergency admission by using data from the nationwide Japan Trauma Data Bank. Both death in the ER and death upon hospital discharge were considerably lower during the day than at night after controlling for confounding factors. It is interesting to note that the same country data might provide different results at the national, regional, and institutional levels. Our observations agree with Hirose’s findings in Japan and others, such as the report by Barbosa et al., USA [29], who reported increased mortality during nights but not on weekends, similar to this cohort.

The increased incidence of significant trauma documented outside of working hours may place additional strain on the limited resources of the night and weekend shifts, perhaps leading to a negative impact on major trauma outcomes. Night shift employment has been linked to exhaustion, poor sleep quality, mood swings, irritability, and health problems; additionally, these differences can be partially explained by less staffing [8,19,20,23]. Previous research has shown that physicians who work successive night shifts have a far more considerable drop in cognitive function than those who work day shifts [7,30,31,32]. As a result, doctor errors may be more frequent during off-working hours [33]. These findings imply that the reduction in cognitive abilities of emergency and other medical professionals working at night may be the cause of poor patient prognosis.

Nevertheless, our findings show that medical care providers should be worried about the possibility of increased risks during off-working hours. Our study noted that trauma patients hospitalized after off-working hours were more likely to be younger and more severely injured than those admitted during regular working hours. Previous reports have indicated similar patterns [2,27,34,35]. This could be due to the tendency of younger people to venture out late at night or on weekends, where they may be engaged in significant traffic accidents, resulting in injuries [21,36].

Overall, the truth regarding off-working hours mortality remains unclear, and it is a multifaceted phenomenon that expresses itself in different ways at different setups and among different disease processes. Therefore, the challenge for healthcare policymakers is to determine the cost-effectiveness of improved staffing in significant trauma centers [12,30].

The weekend effect has been observed in many distinct patient groups and has been termed as “pervasive” [37,38,39]. In the current study, the ISS was significantly higher during weekends than during weekdays (*p* = 0.01), but other clinical characteristics and care and outcome parameters did not differ between the two groups. No significant differences were noticed in the LOS (ICU and hospital) and the mortality.

Our study found no evidence of a weekend effect in injured patients for any significant trauma condition, although there was an increase in admissions, mirroring the recognized epidemiology of trauma. Our findings are in keeping with a prior study [3] and other earlier investigations [24,40], which found no evidence of trauma weekend mortality. A more recent national-level Japanese study by Hirose et al. (2020, Japan) [28] evaluated the outcomes of trauma patients according to the time of day or day of the week of emergency admission. They also found that weekdays/weekends were not associated with the outcomes/mortality of these patients. Furthermore, our results are in keeping with studies from other large regional hospitals, which found no increase in mortality for trauma patients on weekends [39,41,42].

Some previous studies contradict our findings. According to an American study, patients admitted at night are 1.2 times more likely to die than those admitted during business hours [32]. Another study by Pauls et al., who carried out a systematic review of 97 English studies, examined inpatient mortality for the patients admitted on weekends vs. those admitted during the weekdays. He found a significantly higher overall mortality regardless of differences in staffing, procedure rates, and delays, and disease severity [43]. A recent US report by Sharp et al. found higher mortality for all sick patients admitted during the weekends [44].

Weekend admissions may have several characteristics of care that disfavor patients. Previous research has suggested that the weekend effect can be explained by reduced staffing levels [45], the use of temporary clinical staff [46], reduced availability of investigations, diagnostics, and procedures [47], and decreased access to evaluate cases by senior staff, all of which may have a negative impact on the outcome. Why could a “weekend effect” for trauma patients not be found in the current and prior studies? The reason could be attributed to the fact that our hospital is a level 1 trauma center in Qatar, equivalent in terms of care to trauma centers in the United States of America and the United Kingdom. Our hospital has trauma systems to treat emergency trauma patients, and medical resources are available around the clock. Therefore, it is well-equipped to deliver a consistent trauma service and, as a result, always achieves comparable outcomes. As a result, even on weekends/holidays or at night, emergency trauma patients could be treated as usual. Due to these circumstances, we were most likely unable to detect the weekend impact in the current study. Our findings, however, cannot be applied to all level 1 trauma facilities. This finding should be seen in the light of previous research that has found weekend impacts in other emergency groups.

Comparing data requires an understanding of the existing literature background. The health system, case mix, and other comparable characteristics, for example, vary substantially between the different locations and nations. The study by Meacock et al. [48] describes the observed higher mortality among admitted patients during weekends in the UK as a statistical flaw as fewer and sicker patients (selection effect) are admitted. This raises the mortality rate, though the actual toll of death does not change, and the potential for prevention is not valid. An earlier report from the UK (across England and Wales) shed light on the increase in mortality weekend effects. It found that effects are mainly on the medical diseases concerning age-related influences. It showed a close relation to age-related conditions, namely cancer and, to a lesser extent, circulatory and similar conditions, and the statistical artifact effect of less admission on the weekend demonstrated by Roberts et al. [49]. Another dimension for data heterogeneity is defining weekends and night shifts as off-working hours; Barbosa et al. [29] defined shifts (day 08:00 to 17:59 h and night 18:00 h to 7:59 h) and weekends (between Friday 18:00 h until 07:59 h on Monday), for example. He showed increased surgically operated trauma patients on weekends, with higher mortality with three independent predictors (age, red risk rating, and night admission, not at the weekends). These findings could also be due to climate differences.

Weather is thought to play a role in seasonal changes in trauma admissions. Local climatic conditions are considered to impact the number of trauma admissions. In the published research, specific weather patterns, such as warm, sunny days or heavy rainfall, have been related to a greater frequency of trauma admissions [15,50,51,52]. Regarding the potential influence of weather on trauma admissions and outcomes, no significant seasonal variation was observed in terms of admissions at the trauma center. Except for a relative drop in admissions during summer, other variables, such as the injury severity and mortality, did not reflect any statistical difference for the intervals of Qatar’s different seasons.

The cause of the peak in trauma admissions observed during autumn (September to November) and winter (November–February in particular) could be attributed to the fact that these months have the optimal temperature that could increase outdoor activity and are likely to contribute to trauma. We noted a marginal and slight decrease in admissions during the summer months (June–August), possibly due to extreme heat and comparatively higher temperatures in the Middle East region during these months. We hypothesized that the decrease in admissions observed in our data was because the population in Qatar is more likely to alter their normal activities in relation to hot weather and is more likely to remain indoors.

Available literature from different world regions reported a variation in seasonal effects on trauma admissions and outcomes. Few studies have reported an impact on admission rather than outcomes in summer daytime [15,51]. Bundi et al. [52] showed that a higher daytime temperature, prolonged sunshine, and lower humidity accounted for 30% of the trauma admissions in Switzerland’s lowland. However, it did not influence the outcome, as it did in our study [52]. Pape-Kohler et al. [4] conducted a 10-year retrospective study in Germany to determine the external factors that could affect the incidence and outcomes, namely hospital mortality. The months of summer, i.e., June and July, showed the highest incidence. At the same time, the outcome did not show any significant changes [4]. Overall, we did not observe any seasonal effect on the trauma admissions and outcomes in Qatar. Moreover, our findings do not fit with the overall patterns outlined in the literature. There is currently not enough literature to compare our findings on injury severity patterns and outcomes by season.

In our study, the multivariate logistic regression analysis showed that age, injury severity score, head injury, and ED presentation during off-working hours were found to be the significant predictors of mortality. The “off-hours” operations are more complex and require unusual logistics. Failure to operate fully, with some services unavailable, results in a delay in execution and challenging logistics service. Another factor that may have an important association is the fatigue of professionals at night. Additionally, the disruption of the circadian cycle can change and influence the procedures adopted [24,27,53]. This study, conducted in the only level 1 tertiary care trauma hospital, is consistent with findings in the literature about the mortality of patients admitted in different shifts, with higher mortality in patients admitted during the “off-hours” time [32].

This study pinpoints that the relationship between the organizational level and increased mortality during “off hours” should be acknowledged and used as an important indication to optimize care procedures and modify these processes throughout time. It is critical to recognize that the period “off hours” has organizational features that must be controlled to solve these peculiarities, which are not identical to daytime shifts to improve services. Clinicians/surgeons should be aware that such hazards can increase during off-hours and can have adverse outcomes.

This study is not without limitations. First, the retrospective design of the current study is one of its primary limitations; it cannot exclude the possibility of other confounding factors that were not analyzed. The unmeasured confounding factors could have altered the relationship between trauma patient outcomes and the time of day or week or on weekends. Second, our findings may not apply to other institutions because our study population was small compared to other studies and was conducted in a single level 1 trauma center with optimal staffing and diagnostic and therapeutic services. Future studies may be conducted in collaboration with other specialized trauma centers. Third, our analysis did not take public holidays into account. Fourth, because this registry does not include trauma patients who died before arriving at the hospital, we could not compare them to trauma patients who were alive when they arrived. Finally, data were collected during a short period in a single center. There was no calculation made for the sample size, which may restrict the study’s external validity.

The strength of this study is that this is the first report from the Arab Middle East region to address the impact of off-working hours admission on trauma management and its relationship between trauma emergency outcomes. The current study adds to the recent debate surrounding case volume, hospital volume, and outcomes related to the time of day or availability of services. Our findings can be used by specialists, nurses, hospital administrators, and health policymakers to improve the quality and consistency of medical treatment. Awareness about the timing of trauma and its impact will be helpful in injury prevention plans and the preparation for mass gathering events such as the upcoming FIFA 2022 World Cup in Qatar.

## 5. Conclusions

The off-working hours’ admission showed an apparent demographic effect, involved mechanisms, injury severity, trauma activations, and outcomes, especially the mortality rate. The weekend admissions showed a close pattern of effect on gender, severity, trauma activation, and hospital resource utilization, but not the outcome, which is an interesting observation. The different seasons of the year did not show any difference regarding the injury patterns, management, or outcome measures, another remarkable observation but with lower rates of admission during the summer break. Future research should be conducted to determine which datasets and hospital services display a weekend effect to determine whether some patients genuinely have worse outcomes when admitted during weekends.

## Figures and Tables

**Figure 1 ijerph-18-08542-f001:**
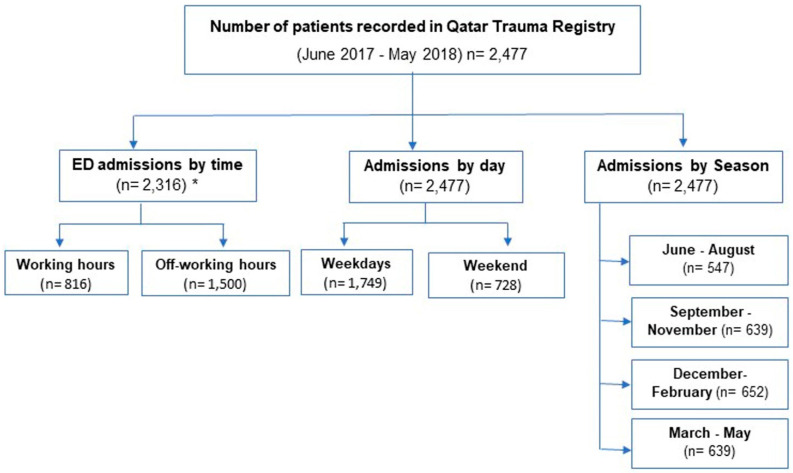
Study schematic (* time of ED admission was missing in 161 cases).

**Figure 2 ijerph-18-08542-f002:**
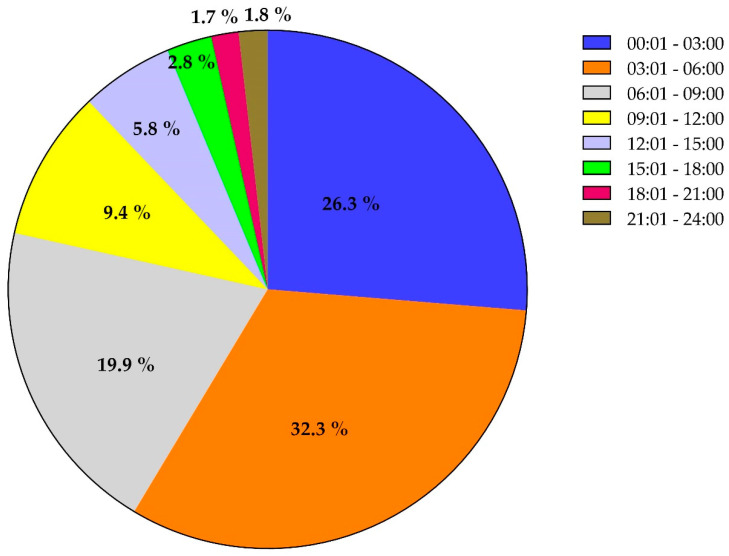
Analysis of time to ED presentation during the one-year period. The majority (almost 58%) of the patients were presented to the ED during 03:01–6:00 h (32.3%) and 00:01–3:00 h (26.3%). This was followed by time windows of 06:00–09:00 h (19.9%) and 09:01–12:00 h (9.4%), respectively.

**Figure 3 ijerph-18-08542-f003:**
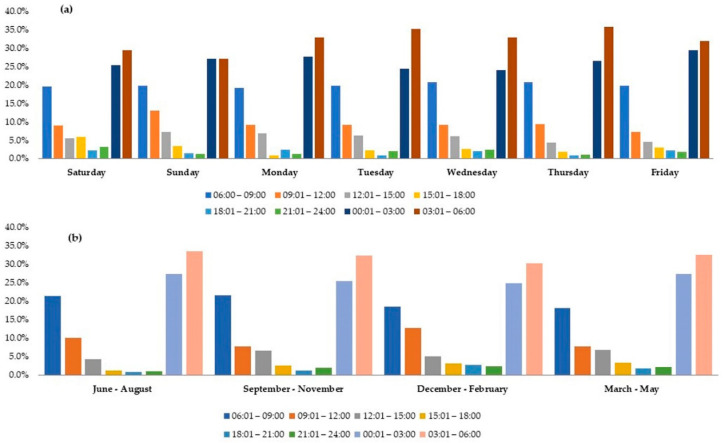
Analysis of time to ED presentation by weekdays vs. weekends and seasons of the year. Weekday vs. wed trends showed most of the patients being presented to ED on weekends and a significant percentage of patients presented during early morning hours (**a**). The seasonal breakdown of ED admissions is shown in (**b**) (June–August, September–November, December–February, and March–May).

**Table 1 ijerph-18-08542-t001:** Comparison of demographics and outcome by time in a day.

	Working Hours (06:01–15:00 h) *n* = 816	Off-Working Hours (15:01–06:00 h) *n* = 1500	Odd Ratio (95% CI)	*p*-Value
**Age** (**years**)	33.5 ± 15.3	28.8 ± 15.5	-	0.001
≤15	74 (9.2%)	257 (17.9%)	3.91 (2.27–6.71)	0.001 for all
16–64	698 (86.4%)	1143 (79.8%)	1.84 (1.13–2.99)	
≥65	36 (4.5%)	32 (2.2%)	1 (Ref)	
**Males**	708 (86.8%)	1250 (83.3%)	1.31 (1.03–1.67)	0.02 for all
**Females**	108 (13.2%)	250 (16.7%)	1 (Ref)	
**Mechanism of injury**				0.001 for all
RTI	435 (53.3%)	713 (47.5%)	0.61 (0.48–0.76)	
Fall	241 (29.5%)	410 (27.3%)	0.63 (0.49–0.81)	
Others	140 (17.2%)	377 (25.1%)	1 (Ref)	
**Priority of activation**				
P1	44 (5.4%)	339 (22.6%)	3.91 (2.68–5.71)	0.001 for all
P2	642 (78.7%)	905 (60.3%)	0.72 (0.56–0.90)	
Others	130 (15.9%)	256 (17.1%)	1 (Ref)	
**Response time**	7 (1–56)	7 (1–132)	-	0.23
**Scene Time ≤ 20 min.** (***n* = 1686**)	326 (52.1%)	512 (48.3%)	1 (Ref)	0.13 for all
**Scene Time > 20 min.**	300 (47.9%)	548 (51.7%)	1.16 (0.95–1.42)	
**Prehospital time****≤ 60 min.** (***n* = 1736**)	193 (30.0%)	381 (34.9%)	1 (Ref)	0.03 for all
**Pre-hospital time** **> 60 min.**	450 (70.0%)	712 (65.1%)	0.80 (0.65–0.99)	
**Blood Transfusion**	71 (8.7%)	256 (17.1%)	2.16 (1.64–2.85)	0.001
**MTP Activated**	6 (0.7%)	80 (5.3%)	7.60 (3.30–17.51)	0.001
**Surgery within 24 h**	144 (17.6%)	320 (21.3%)	1.27 (1.02–1.57)	0.03
**Intubation**	41 (5.0%)	366 (24.4%)	6.10 (4.36–8.53)	0.001
**Exploratory laparotomy**	20 (2.5%)	78 (5.2%)	2.18 (1.33–3.60)	0.002
**ISS** (**mean ± SD**)	11.2 ± 7.8	14.8 ± 11.6	-	0.001
**ICU LOS** (**days**)	3 (1–76)	4 (1–102)	-	0.001
**Hospital LOS** (**days**) *****	4 (1–120)	4 (1–166)	-	0.70
**Transferred to rehabilitation**	31 (3.8%)	93 (6.2%)	1.67 (1.10–2.54)	0.01
**Mortality**	6 (0.7%)	145 (9.7%)	14.45 (6.35–32.84)	0.001

*: LOS = length of stay.

**Table 2 ijerph-18-08542-t002:** Comparison of demographics and outcome by day of the week.

	Weekdays (*n* = 1749)	Weekend * (*n* = 728)	Odd Ratio (95% CI)	*p*-Value
**Age** (**years**)	31.1 ± 15.7	30.4 ± 16.3	-	0.37
≤15	228 (13.5%)	115 (16.5%)	1.31 (0.76–2.24)	0.16 for all
16–64	1405 (83.1%)	560 (80.3%)	1.03 (0.62–1.71)	
≥65	57 (3.4%)	22 (3.2%)	1 (Ref)	
**Males**	1493 (85.4%)	596 (81.9%)	1.29 (1.03–1.63)	0.02 for all
**Females**	256 (14.6%)	132 (18.1%)	1 (Ref)	
**Mechanism of injury**				0.12 for all
RTI	844 (48.3%)	371 (51.0%)	0.99 (0.80–1.23)	
Fall	516 (29.5%)	185 (25.4%)	0.81 (0.63–1.04)	
Others	389 (22.2%)	172 (23.6%)	1 (Ref)	
**Priority of activation**				
P1	257 (14.7%)	139 (19.1%)	1.63 (1.21–2.19)	0.01 for all
P2	1164 (66.6%)	480 (65.9%)	1.24 (0.97–1.58)	
Others	328 (18.8%)	109 (15.0%)	1 (Ref)	
**Response time**	7 (1–89)	7 (1–132)	-	0.83
**Scene Time ≤ 20 min.** (***n* = 1806**)	641 (51.0%)	258 (46.9%)	1 (Ref)	0.10 for all
**Scene Time > 20 min.**	615 (49.0%)	292 (53.1%)	1.18 (0.97–1.44)	
**Prehospital time****≤ 60 min.** (***n* = 1852**)	400 (31.0%)	208 (37.0%)	1 (Ref)	0.01 for all
**Pre-hospital time** **> 60 min.**	890 (69.0%)	354 (63.0%)	0.77 (0.62–0.94)	
**Blood Transfusion**	232 (13.3%)	108 (14.8%)	1.14 (0.89–1.46)	0.30
**MTP Activated**	56 (3.2%)	32 (4.4%)	1.39 (0.89–2.17)	0.14
**Surgery within 24 h**	329 (18.8%)	145 (19.9%)	1.07 (0.86–1.34)	0.52
**Intubation**	275 (15.7%)	152 (20.9%)	1.41 (1.14–1.76)	0.002
**Exploratory lap.**	73 (4.2%)	27 (3.7%)	0.88 (0.56–1.39)	0.59
**ISS** (**mean ± SD**)	12.8 ± 9.8	14.1 ± 11.1	-	0.01
**ICU LOS** (**days**)	3 (1–102)	4 (1–63)	-	0.13
**Hospital LOS** (**days**)	4 (1–166)	4 (1–130)	-	0.12
**Transferred to rehabilitation**	89 (5.1%)	42 (5.8%)	1.14 (0.78–1.67)	0.49
**Mortality**	115 (6.6%)	54 (7.4%)	1.13 (0.81–1.59)	0.44

* Friday and Saturday.

**Table 3 ijerph-18-08542-t003:** Comparison of demographics and outcomes by season.

Variable	June–August (*n* = 547)	September–November (*n* = 639)	December–February (*n* = 652)	March–May (*n* = 639)	*p*-Value
**Age (years)**	31.3 ± 16.6	30.0 ± 14.4	30.6 ± 15.9	31.6 ± 16.4	0.29
≤15	78 (14.8%)	77 (12.4%)	91 (14.7%)	97 (15.6%)	0.54 for all
16–64	435 (82.5%)	527 (84.7%)	505 (81.7%)	499 (80.5%)	
≥65	14 (2.7%)	18 (2.9%)	22 (3.6%)	24 (3.9%)	
**Males**	474 (86.7%)	535 (83.7%)	552 (84.7%)	529 (82.8%)	0.30 for all
**Females**	73 (13.3%)	104 (16.3%)	100 (15.3%)	110 (17.2%)	
**Mechanism of injury**					0.92 for all
RTI	261 (47.7%)	317 (49.6%)	319 (48.9%)	319 (49.9%)	
Fall	156 (28.5%)	173 (27.1%)	193 (29.6%)	178 (27.9%)	
Others	130 (23.8%)	149 (23.3%)	140 (21.5%)	142 (22.2%)	
**Priority of activation**					
P1	89 (16.3%)	96 (15.0%)	108 (16.6%)	103 (16.1%)	0.60 for all
P2	373 (68.2%)	436 (68.2%)	423 (64.9%)	413 (64.6%)	
Others	85 (15.5%)	107 (16.7%)	121 (18.6%)	123 (19.2%)	
**Response time**	7 (1–132)	7 (1–89)	7 (1–41)	7 (1–55)	0.49
**Scene Time ≤ 20 min. (** ***n*** ** = 1807)**	169 (48.4%)	218 (54.1%)	267 (50.1%)	247 (47.3%)	0.20 for all
**Scene Time > 20 min.**	180 (51.6%)	185 (45.9%)	266 (49.9%)	275 (52.7%)	
**Prehospital time ≤ 60 min. (** ***n*** ** = 1853)**	118 (32.4%)	143 (34.9%)	178 (32.5%)	169 (31.8%)	0.77 for all
**Pre-hospital time > 60 min.**	246 (67.6%)	267 (65.1%)	369 (67.5%)	363 (68.2%)	
**Blood Transfusion**	70 (12.8%)	85 (13.3%)	93 (14.3%)	92 (14.4%)	0.82
**MTP Activated**	17 (3.1%)	19 (3.0%)	23 (3.5%)	29 (4.5%)	0.43
**Surgery within 24 h**	118 (21.6%)	124 (19.4%)	109 (16.7%)	123 (19.2%)	0.20
**Intubation**	87 (15.2%)	98 (15.3%)	121 (18.6%)	121 (18.9%)	0.22
**Exploratory lap.**	26 (4.8%)	23 (3.6%)	22 (3.4%)	29 (4.5%)	0.53
**ISS (mean ± SD)**	12.6 ± 10.1	13.3 ± 11.2	13.6 ± 9.9	13.3 ± 9.9	0.59
**ICU LOS (days)**	3 (1–76)	4 (1–62)	4.5 (1–76)	4 (1–102)	0.19
**Hospital LOS (days) **	4 (1–129)	4 (1–82)	4 (1–130)	4 (1–166)	0.70
**Transferred to rehabilitation**	21 (3.8%)	35 (5.5%)	41 (6.3%)	34 (5.3%)	0.30
**Mortality**	30 (5.5%)	42 (6.6%)	53 (8.1%)	44 (6.9%)	0.34

**Table 4 ijerph-18-08542-t004:** Comparison between survivors and non-survivors among trauma patients.

Variables	Survivors (*n* = 2308)	Non-Survivors (*n* = 169)	OR (95% CI)	*p*-Value
**Age (years)**	30.8 ± 15.8	34.4 ± 16.0	-	0.03
**Male gender**	1927 (83.5%)	162 (95.9%)	0.219 (0.102–0.469)	0.001
**Mechanism of injury**				
RTI	1092 (47.3%)	123 (72.8%)	2.76 (1.73–4.39)	0.001
Fall	677 (29.3%)	24 (14.2%)	0.86 (0.47–1.58)	0.63
Others	539 (23.4%)	22 (13.0%)	1 (Ref)	1 (Ref)
Prehospital time ≤ 60 min	558 (32.7%)	50 (34.0%)	1 (Ref)	
Pre-hospital time > 60 min	1147 (67.3%)	97 (66.0%)	0.94 (0.66–1.34)	0.75
Priority of activation				
P1	294 (12.7%)	102 (60.4%)	2.06 (1.45–2.92)	0.001
P2	1640 (71.0%)	4 (2.4%)	0.01 (0.005–0.04)	0.001
Others	374 (16.2%)	63 (37.3%)	1 (Ref)	
**MTP activation**	50 (2.2%)	38 (22.5%)	13.10 (8.29–20.69)	0.001
**Surgery within 24** **h**	447 (19.4%)	27 (16.0%)	0.79 (0.52–1.21)	0.27
**Injury severity score**	12.3 ± 9.1	22.9 ± 15.7	-	0.001
**Head injury**	504 (21.8%)	104 (61.5%)	5.72 (4.13–7.92)	0.001
**Admission on Working h ***	810 (37.4%)	6 (4.0%)	1 (Ref)	
**Admission Off-working h**	1353 (62.6%)	146 (96.0%)	14.46 (6.36–32.88)	0.001
**Weekday admission**	1634 (70.8%)	115 (68.0%)	1 (Ref)	0.44
**Weekend admission**	674 (29.2%)	54 (32.0%)	1.14 (0.81–1.59)	
**Season admission**				
June–August	517 (22.4%)	30 (17.8%)	1 (Ref)	0.34
September–November	595 (25.8%)	42 (24.9%)	1.21 (0.75–1.98)	0.42
December–February	599 (26.0%)	53 (31.4%)	1.52 (0.95–2.42)	0.07
March–May	595 (25.8%)	44 (26.0%)	1.27 (0.78–2.05)	0.31

* Injury by hours is available in 2314 cases; pre-hospital time available in 1852 cases.

**Table 5 ijerph-18-08542-t005:** Multivariate regression analysis for the predictors of mortality.

Variables	Odd Ratio	95% Confidence Interval		*p*-Value
		Lower	Upper	
Age	1.020	1.004	1.035	0.01
Males	0.549	0.188	1.598	0.27
Injury severity Score	1.042	1.018	1.067	0.001
Head injury	4.152	2.310	7.464	0.001
Massive transfusion protocol activation	4.192	2.127	8.262	0.001
Off-working h	6.329	2.673	14.985	0.001

## Data Availability

All data related to this study has been given in the results section, figures, and tables.

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
