# Peer review of "The Patterns and Impact of Off-Working Hours, Weekends and Seasonal Admissions of Patients with Major Trauma in a Level 1 Trauma Center"

_ijerph, 2021, doi:10.3390/ijerph18168542_

Round 1

Reviewer 1 Report

In their retrospective case-control study, Abdelrahman et al. examined differences between trauma patients admitted to a Level I center in Qatar during core or on-call hours, and on weekdays or weekends.

The manuscript was evaluated by using a critical appraisal protocol (CASP checklist) and professional judgment.

Major comments

  1. A central part of your work is the comparison with regard to the endpoint "mortality". Your study cannot provide any reliable results, as the statistical methodology does not allow this. As you correctly write yourself, the cohorts differ in essential characteristics. These include ISS, age, gender distribution, mechanism of injury, intubation etc. Thus, a mortality difference could easily be explained by e.g. the higher ISS. Thus, you have to exclude possible confounders. However, this is only possible by e.g. a multivariable regression analysis with preceding univariable analyses. Alternatively, you can apply the TRISS or RISC II methodology and use one of these scores for the correction. Besides, it is international practice to report standardized mortality ratios. 
  2. Unfortunately, the DIscussion cannot be evaluated because you wrote it against the background of supposed mortality differences in your data. Since this has not been demonstrated due to the lack of multivariable regression, you cannot make this statement and have the discussion that way.
  3. Please provide odds ratios and confidence intervals.
  4. Please create a flow-chart on the inclusion and exclusion criteria (PRISMA statement).
  5. Your abstract is way too long. According to the "instructions for authors" of the journal, the abstract should not exceed 200 words.
  6. Major well-established and internationally accepted prognostic parameters are not reported e.g.: Penetrating vs. blunt trauma, shock, GCS, heart rate, INR, AIS head/thorax/abdomen/extremities, ASA score, pupil size, CPR, respiratory rate. These are part of the main prognostic scores (TRISS, RISC II) for a reason. It is mandatory to provide these data!

Minor comments

  1. Please decide on a term for the core working hours or the on-call hours. You can choose to write "off-hours", "off-working hours", "out off-hours", "weeknight". Please provide a clear definition in the methods section and use a consistent term.
  2. What do you mean by "pre-hospital level of activation"? A definition for P1 or P2 is missing.
  3. You have specified time periods that you are comparing (e.g. 6:01-18:00). Please use the times also for your illustrations and do not break them down in too much detail.
  4. Your introduction is very long, should be shortened to half the length, and should be precise in pointing out the topic. Please also include a PICO statement.
  5. Has your study been registered in an international study registry a priori?
  6. P. 3 ll. 108-110 "Could the trauma system [in Quatar] protect[...]"
  7. P. 3 l. 115 This is a retrospective study. The reference that data were collected prospectively (which is always the case for treatment-relevant data at hospital admission) confuses the reader and should be deleted.
  8. P. 3 l. 122 Is the trauma surgeon a specialist?
  9. Your definitions on times are confusing. Please provide more clarity here. E.G.: 
    Core working hours were defined as working hours from 6 am to 3 pm. On-call hours were defined as from 3:01 p.m. to 5:59 a.m. Weekends were defined as from Friday X p.m. to Saturday X p.m.
  10. The subheadings in the discussion do not conform to the format requirements of the journal.
  11. Table 4 is redundant.
  12. P.14 l. 465 What does NICU mean? Please provide abbreviations for all abbreviations in a list of abbreviations.
  13. Their limitations are much too long. Moreover, from p. 14 l.482 you start again with the discussion.
  14. Important definitions are missing in the methods section, e.g. "Blood transfusion", "response time", "RTI" etc.
  15. Pg 10 l.288 You begin your discussion with outcome after myocardial infarction. Here you compare two fundamentally different disease entities. You should delete the relevant sentences.

Author Response

R1:

In their retrospective case-control study, Abdelrahman et al. examined differences between trauma patients admitted to a Level I center in Qatar during core or on-call hours, and on weekdays or weekends.

The manuscript was evaluated by using a critical appraisal protocol (CASP checklist) and professional judgment.

Major comments

  1. A central part of your work is the comparison with regard to the endpoint "mortality". Your study cannot provide any reliable results, as the statistical methodology does not allow this. As you correctly write yourself, the cohorts differ in essential characteristics. These include ISS, age, gender distribution, mechanism of injury, intubation etc. Thus, a mortality difference could easily be explained by e.g. the higher ISS. Thus, you have to exclude possible confounders. However, this is only possible by e.g. a multivariable regression analysis with preceding univariable analyses. Alternatively, you can apply the TRISS or RISC II methodology and use one of these scores for the correction. Besides, it is international practice to report standardized mortality ratios. 

Reply: We would like to thank the reviewer for the insightful comments. Regarding this study we don’t think that central part of our work was comparison of trauma patients with regard to the mortality. We wanted to explore the patterns and effects of off-hours, weekends, and seasonal effects on trauma patients ( occurrence , pattern and severity) for which mortality is one of the observed parameters. As per suggestion we have incorporated 2 new tables ; one for comparison between survivors and non survivors (tab 4) and one for the results of multivariate regression analysis (table 5).

  1. Unfortunately, the Discussion cannot be evaluated because you wrote it against the background of supposed mortality differences in your data. Since this has not been demonstrated due to the lack of multivariable regression, you cannot make this statement and have the discussion that way.

Reply: We have modified our manuscript by adding the results of multivariate analysis.

  1. Please provide odds ratios and confidence intervals.

Reply: Done

  1. Please create a flow-chart on the inclusion and exclusion criteria (PRISMA statement).

Reply: We have retrospectively reviewed data for all the trauma patients recorded in Qatar Trauma Registry from June 2017 - May 2018. We have included the study schematic as figure1 in the manuscript.

  1. Your abstract is way too long. According to the "instructions for authors" of the journal, the abstract should not exceed 200 words.

Reply: As per the journal requirement we have shortened the abstract.

  1. Major well-established and internationally accepted prognostic parameters are not reported e.g.: Penetrating vs. blunt trauma, shock, GCS, heart rate, INR, AIS head/thorax/abdomen/extremities, ASA score, pupil size, CPR, respiratory rate. These are part of the main prognostic scores (TRISS, RISC II) for a reason. It is mandatory to provide these data!

Reply: We agree with the points related to prognostic parameters by the reviewer. However, our manuscript is not about prognosis, it is mainly to describe the pattern and impact of different timings of trauma . no difference in incidence and severity exist in different aspects as the manuscript describe it but there was no intension to prognosticate trauma patients in general nor specific region sub-analysis.also as it is a retrospective study, some of such variables are not captured

Minor comments

  1. Please decide on a term for the core working hours or the on-call hours. You can choose to write "off-hours", "off-working hours", "out off-hours", "weeknight". Please provide a clear definition in the methods section and use a consistent term.

Reply: We have used "working and off-working hours" consistently throughout the manuscript and it has been defined in the methods section.

  1. What do you mean by "pre-hospital level of activation"? A definition for P1 or P2 is missing.

Reply: The Standard trauma team activation is based on the EMS dispatch system which is on-call for service code into patient’s priority or P1 (emergency), P2 (urgent) and others (routine). The same has been incorporated in the method section.

  1. You have specified time periods that you are comparing (e.g. 6:01-18:00). Please use the times also for your illustrations and do not break them down in too much detail.

Reply: It has been illustrated in Figure 2.

  1. Your introduction is very long, should be shortened to half the length, and should be precise in pointing out the topic. Please also include a PICO statement.

Reply: Done. Introduction is almost one and quarter page,

  1. Has your study been registered in an international study registry a priori?

Reply: The study has been registered and approved from the medical research center and institutional review board of Hamad Medical Corporation, Doha, Qatar (MRC-01-18-432). Considering this as an retrospective observational study, it is not registered with an international body.  

  1. P. 3 ll. 108-110 "Could the trauma system [in Quatar] protect[...]"

Reply: Done

  1. P. 3 l. 115 This is a retrospective study. The reference that data were collected prospectively (which is always the case for treatment-relevant data at hospital admission) confuses the reader and should be deleted.

Reply: Done

  1. P. 3 l. 122 Is the trauma surgeon a specialist?

Reply: Yes, specialist board certified surgeons.

  1. Your definitions on times are confusing. Please provide more clarity here. E.G.: 
    Core working hours were defined as working hours from 6 am to 3 pm. On-call hours were defined as from 3:01 p.m. to 5:59 a.m. Weekends were defined as from Friday X p.m. to Saturday X p.m.

Reply: We have modified the manuscript and state the definitions clearly in the methodology section.

  1. The subheadings in the discussion do not conform to the format requirements of the journal.

Reply: The subheadings in the discussion part has been deleted to meet the format requirements of the journal.

  1. Table 4 is redundant.

Reply: Table 4 has been deleted.

  1. P.14 l. 465 What does NICU mean? Please provide abbreviations for all abbreviations in a list of abbreviations.

Reply: NICU has been written inadvertently, which has been modified in the text.

  1. Their limitations are much too long. Moreover, from p. 14 l.482 you start again with the discussion.

Reply: The discussion part including limitations has been modified and shortened.

  1. Important definitions are missing in the methods section, e.g. "Blood transfusion", "response time", "RTI" etc.

Reply: The same has been corrected in the methods.

  1. Pg 10 l.288 You begin your discussion with outcome after myocardial infarction. Here you compare two fundamentally different disease entities. You should delete the relevant sentences.

Reply: The same has been corrected in the methods.

Reviewer 2 Report

  1. It is a very interesting study attempting to show the effects of different time periods of a day, weekdays, weekends on the trauma incidence and on a list of outcomes in a high standard hospital in the Middle East. As most of these studies came from north America and Japan as quoted in the manuscript, readers were eager and happy to know if the results of this study had a regional characteristic. To make this article more interesting to readers, I would suggest the authors to provide more information of patient characteristics, type of trauma, seasonal changes, etc., I think these should reveal regional characteristics.
  2. English writing should be improved. I would suggest that the manuscript should be sent for English editing. For example, number should appear as 2,477 instead of 2477, and abbreviation should be clearly defined at the first time it appeared in the manuscript.
  3. As authors quoted many previous studies that they had adjusted for confounders, I would suggest that adjusting for confounders in this study is necessary.
  4. Holidays were excluded in this study. What are the reasons?
  5. There were different terms, such as, working hours, outside of work hour, regular working hours, off hours, after hours, etc., I would say that too many terms about time intervals would confuse readers.
  6. Discussion is a place where we could discuss interesting findings of the study. Comparing or quoting previous report is necessary but discussion of your findings is more important. Take for an example, intubation rate was significantly higher in the off hours group, reader would be happy to hear detailed explanation. The manuscript revealed many interesting findings, nevertheless, few was discussed.
  7. Weekend effect was not found in the study because your hospital is a Level I trauma center in Qatar, even on weekends/holidays or at night, emergency trauma patients could be treated as usual. I would consider that that there should be no difference in the outcome of patient no matter in working hours or in off hours.

Author Response

R2:

  1. It is a very interesting study attempting to show the effects of different time periods of a day, weekdays, weekends on the trauma incidence and on a list of outcomes in a high standard hospital in the Middle East. As most of these studies came from north America and Japan as quoted in the manuscript, readers were eager and happy to know if the results of this study had a regional characteristic. To make this article more interesting to readers, I would suggest the authors to provide more information of patient characteristics, type of trauma, seasonal changes, etc., I think these should reveal regional characteristics.

Reply: We have added more information about patient characteristics at the beginning of the result section and accordingly manuscript has been modified.

  1. English writing should be improved. I would suggest that the manuscript should be sent for English editing. For example, number should appear as 2,477 instead of 2477, and abbreviation should be clearly defined at the first time it appeared in the manuscript.

Reply: Done

  1. As authors quoted many previous studies that they had adjusted for confounders, I would suggest that adjusting for confounders in this study is necessary.

Reply: We would like to thank the reviewer for the insightful comments. As per suggestion we have incorporated the new set of results in which we have performed multivariate regression analysis after adjusting for the potential confounders (table 5).

  1. Holidays were excluded in this study. What are the reasons?

Reply: “Public holidays were not counted as weekend or weekday admissions and were excluded from the analyses of weekend mortality effects”. The registry does not capture the holidays and it was not considered.

  1. There were different terms, such as, working hours, outside of work hour, regular working hours, off hours, after hours, etc., I would say that too many terms about time intervals would confuse readers.

Reply: We have used "working and off-working hours" consistently throughout the manuscript and it has been defined in the methods section.

  1. Discussion is a place where we could discuss interesting findings of the study. Comparing or quoting previous report is necessary but discussion of your findings is more important. Take for an example, intubation rate was significantly higher in the off hours group, reader would be happy to hear detailed explanation. The manuscript revealed many interesting findings, nevertheless, few was discussed.

Reply: We have revisited the discussion part and have made changes to it to address the reviewer’s comments.

  1. Weekend effect was not found in the study because your hospital is a Level I trauma center in Qatar, even on weekends/holidays or at night, emergency trauma patients could be treated as usual. I would consider that that there should be no difference in the outcome of patient no matter in working hours or in off hours.

Reply: Though the argument that in level one trauma center the setup and arrangement compensate for the available care related resources the severity and the incidence of injury differ and injury related rather than care related difference plausibly explains the observed difference.

Reviewer 3 Report

This is an interesting study that applied the statistical method to explore the effects of off-hours, weekends, and seasonal effects on major trauma patients. The data are novel and the discussions are concentrated on the topics revealed by the title. The authors make it clear what the aims are in this study, and are able to demonstrate those findings with appropriate methods. I have a few minor comments that need to be addressed:

1) lines 23-24, this sentence is incomplete;

2) lines 139, hrs were used to describe am or pm interchangeably, but I would suggest authors be consistent to use either one of them, and am pm is preferred;

3) lines 167-168, please provide supporting evidence why some data was not used;

4) Figure 1, 00:01-03:00, 03:01-06:00 should be placed at the top of the legend to be aligned with other time slots; similarly, I would suggest Friday and Saturday be placed together as the authors indicated that they are weekends in Figure 2;

5) Table 3, Intubation, some strange number appeared  87 (15.2%)9

6) Not sure why Table 4 is attached at the end of the discussion without too much explanation of this information. I do not think it added significant amount of value towards the end of this study. Please consider removing it or treat it as an attachment;

7) Author contributions: I would suggest authors use the full name instead of abbreviations.

Author Response

R3:

This is an interesting study that applied the statistical method to explore the effects of off-hours, weekends, and seasonal effects on major trauma patients. The data are novel and the discussions are concentrated on the topics revealed by the title. The authors make it clear what the aims are in this study, and are able to demonstrate those findings with appropriate methods. I have a few minor comments that need to be addressed:

1) lines 23-24, this sentence is incomplete;

Reply: The correction has been made in the manuscript.

2) lines 139, hrs were used to describe am or pm interchangeably, but I would suggest authors be consistent to use either one of them, and am pm is preferred;

Reply: we have now modified our manuscript and have taken into account the hours throughout the manuscript consistently.

3) lines 167-168, please provide supporting evidence why some data was not used;

Reply: they were not included in the registry and their data are missing.

4) Figure 1, 00:01-03:00, 03:01-06:00 should be placed at the top of the legend to be aligned with other time slots; similarly, I would suggest Friday and Saturday be placed together as the authors indicated that they are weekends in Figure 2;

Reply: Figure 1 (now Figure 2) has been modified as per suggestion.

5) Table 3, Intubation, some strange number appeared  87 (15.2%)9

Reply: Typo corrected

6) Not sure why Table 4 is attached at the end of the discussion without too much explanation of this information. I do not think it added significant amount of value towards the end of this study. Please consider removing it or treat it as an attachment;

Reply: Table 4 has been removed as per suggestion.

7) Author contributions: I would suggest authors use the full name instead of abbreviations.

Reply: Done

Round 2

Reviewer 1 Report

Most comments were answered satisfactorily.

  1. I would still recommend to add the RISC II or the TRISS to the multivariable regression analysis. This is international standard and should correct the result for most potential confounder.
  2. Is there a specialist present in the ER at night as well? This is rather unusual. If a resident is on ER duty at night and the Specialist is only on call, correction should be made for this as well. Specialist provide an outcome benefit over residents.

Author Response

  1. I would still recommend to add the RISC II or the TRISS to the multivariable regression analysis. This is international standard and should correct the result for most potential confounder.

1- 

Reply: thanks, although the RISC II is of value and used in the German trauma registry, it is more complicated and needs around 13 variables to be calculated. It also includes laboratory results. Most of these variables are not captured in our study and we are not relying on the RISC II as we report our data to the ACS-TQIP in the USA in which RISC II is not part of it. Also it will add colinearity beside the ISS.

TRISS is the probability of survival which is a combination index based on Trauma Score (RTS), Injury Severity Score (ISS), and patient's age. ISS and age are already included in the multivariate analysis. Therefore, we believe that adding TRISS beside age and ISS in the multivariate analysis will not add new result and will have interaction and colinearity effect.

2- Is there a specialist present in the ER at night as well? This is rather unusual. If a resident is on ER duty at night and the Specialist is only on call, correction should be made for this as well. Specialist provide an outcome benefit over residents.

Reply: the trauma team includes the following: consultant on call for T1 activation who is available within 15 min of trauma activation regardless of the timing of the day or week day. Also 2 specialists covering 4 shifts (in-house) over 24 h across the 7 days of the week. The residents are rotating to accompany the specialist in all the shifts. so there is no need to correct for this. We added this comment in the methods section.

Reviewer 2 Report

I have no further comment. 

Author Response

Comments and Suggestions for Authors: I have no further comment.  reply: Thanks